# Effect of SEBS Molecular Structure and Formula Composition on the Performance of SEBS/PP TPE for Automotive Interior Skin

**DOI:** 10.3390/polym15122753

**Published:** 2023-06-20

**Authors:** Shuwen Liu, Jun Qiu, Lili Han, Junping Luan, Xueyan Ma, Wenquan Chen

**Affiliations:** 1College of Material Science and Engineering, Tongji University, Shanghai 200082, China; shuwen-liu@patac.com.cn; 2Pan Asia Technical Automotive Center Co., Ltd., Shanghai 201201, China; 3Shandong Dawn Polymer Co., Ltd., Yantai 265700, China; hanlili022225@foxmail.com (L.H.); mxy@chinadawn.cn (X.M.)

**Keywords:** automotive interior skin, hydrogenated styrene-butadiene-styrene block copolymer, thermoplastic elastomer, wear resistance, orthogonal experiment

## Abstract

The hydrogenated styrene–butadiene–styrene block copolymer (SEBS)/Polypropylene (PP)-blended thermoplastic elastomer (TPE) is an ideal material for automotive interior skin applications due to its excellent elasticity, weather resistance, and environmentally friendly characteristics such as low odor and low volatile organic compounds (VOC). As a thin-wall injection-molded appearance skin product, it requires both high fluidity and good mechanical properties with scratch resistance. To optimize the performance of the SEBS/PP-blended TPE skin material, an orthogonal experiment and other methods were employed to investigate the impact of the formula composition and raw material characteristics, such as the styrene content and molecular structure of SEBS, on the TPE’s final performance. The outcomes revealed that the ratio of SEBS/PP had the most significant influence on the mechanical properties, fluidity, and wear resistance of the final products. The mechanical performance was enhanced by increasing the PP content within a certain range. The degree of sticky touch on the TPE surface was increased as the filling oil content increased, causing the increase in sticky wear and the decrease in abrasion resistance. When the SEBS ratio of high/low styrene content was 30/70, the TPE’s overall performance was excellent. The different proportions of linear/radial SEBS also had a significant effect on the final properties of the TPE. The TPE exhibited the best wear resistance and excellent mechanical properties when the ratio of linear-shaped/star-shaped SEBS was 70/30.

## 1. Introduction

Materials applied in automotive interior are not only required good mechanical properties, but also should exhibit environmentally friendly properties such as low odor, low volatile organic compounds (VOC), and low fogging [1,2,3]. To ensure proper functioning in various climatic conditions, the automotive instrument panel’s soft skin must exhibit excellent stability across a wide range of temperatures and weather conditions, including wet, hot, dry, and UV exposure. Additionally, given that it covers the airbag area, the soft skin must also display excellent toughness to guarantee personal safety during airbag deployment. Traditional automotive interior skins mainly include polyvinyl chloride (PVC) slush skins and thermoplastic polyolefin (TPO) vacuum-molded skins [4]. However, PVC exhibits poor weatherable resistance, and environmentally unfriendly properties such as high odor, VOC emission, fogging, and difficulty to recycle and reuse. Emission of Ester plasticizer of PVC slush skin is harmful to the health of drivers and passengers. In addition, the prolonged process time of PVC slush skin leads to increased energy consumption and elevated process costs. The need for frequent sub-mold replacements further adds to the already high cost of molds for PVC slush skin. Although the vacuum-molded TPO skin has the advantages of low emission behavior such as low odor, low VOC, and low fogging, its production requires complex processes such as extrusion calendaring, vacuum molding, and spray coating. This results in complicated processing technology and extended processing time [5].

One step injection molding process is characterized by higher efficiency and energy saving than slush molding and vacuum molding. Therefore, soft material with proper properties and enough flow-rate for injection of skins with large area and thin-wall would be suitable for replacing traditional interior skins’ materials like PVC and TPO.

Hydrogenated styrene–butadiene–styrene block copolymer (SEBS)/polypropylene (PP)-blended thermoplastic elastomer not only has characteristics of soft touch, but also exhibits low odor, low VOC, and other environmentally friendly characteristics [6]. In addition, TPE exhibits excellent elasticity, aging resistance, weather resistance, wide range temperature stability, and recyclability. The hardness and fluidity of TPE products can be designed within a wide range, making it increasingly popular for automotive interior applications [7,8].

When utilizing the injection molding process to shape the soft skin of the automotive interior, the material requires extremely high fluidity due to the product’s thinness (0.9–1.2 mm) and the vast surface area. Additionally, since the interior skin frequently comes into contact with sharp objects such as nails and keys, it necessitates excellent abrasion and scratch resistance [9]. High fluidity of TPE-S can be achieved by incorporating low-molecular-weight SEBS into the formula or by filling it with a high ratio of low-viscosity paraffin oil to high-molecular-weight SEBS using traditional technology. However, this can lead to a reduction in intermolecular cohesion of SEBS, resulting in poor abrasion and scratch resistance. Such formulations are prone to failure during friction and wear [10]. Additionally, the elastomer material is soft, and when compressed, the contact area is prone to large deformation. This results in the material easily forming a “wrapping” effect on the object experiencing friction. As a result, there is closer contact between the TPE skin and the object experiencing friction, leading to an increase in the friction coefficient, significant generation of frictional heat, and accelerated wear and failure.

For composite materials, the wear mechanisms are mainly abrasive wear, adhesive wear, and fatigue wear. In practical applications, the wear of materials is usually a composite form of two or even three of the three types of wear.

Few people have studied and found suitable methods for balance between high fluidity and enough wear resistance and mechanical properties. In this paper, we utilized the orthogonal experiment [11,12] to explore the impact of the molecular structure of SEBS and various formula factors on the performance of high-flow TPE-S, intended for use in automotive interior soft skin. We evaluated the conventional performance of the material by analyzing its mechanical properties. Additionally, we assessed the fluidity of the material by analyzing its melt flow rate and conducted a Taber abrasion test to examine its wear and scratch resistance. We studied the effect of the PP/SEBS ratio, as well as the content of oil filling in SEBS and PB content, on the product’s performance. We conducted variance analysis on the test results to investigate the significance of each factor on the melt flow rate, tensile strength, tear strength, and wear resistance of the TPE-S product. Finally, we obtained the optimized formula within the test range.

## 2. Materials

SEBS G1651, G1657: American Kraton, commercially available; SEBS YH-602T: China Petroleum & Chemical Corporation (Yueyang, China), commercially available; PP 225 (powder): Zhejiang Hongji Petrochemical Co., Ltd. (Jiaxing, China), commercially available; Paraffin oil KP6030: PetroChina Karamay Petrochemical Co., Ltd. (Karamay, China), commercially available; L Silicone L5-4 (Extra high molecular weight, Mw is about 700,000, Hereinafter refer to as HWSi): Wacker Chemical Group Co., Ltd. (Berghausen, Germany), commercially available.

High-speed mixer: SHR-10A, Guangdong Xieda Machinery Co., Ltd. (Dongguan, China); Twin-screw extruder: SHJ-35, Nanjing Juli Chemical Machinery Co., Ltd. (Nanning, China); Injection molding machine: UN120SK, Yizhimi Precision Machinery Co., Ltd. (Foshan, China); Shore A hardness tester: ZWICK ARMATUREN GMBH (Ulm, Germany); Melt index meter: MF30, CEAST, Italy; Universal tensile testing machine: 3366, Instron Engineering Corporation (Boston, MA, USA); Taber Abrasion Tester: 1755, Taber Industrie (New York, NY, USA),; Cross scraping instrument: 430P-1, ERICHSEN INC. (Ann Arbor, Germany); Two-position imager: SV4030, Dongguan Tianqin Instrument Co., Ltd. (Dongguan, China).

## 3. Results and Discussion

### 3.1. DOE Experiments Design

#### 3.1.1. Determining Factors and Levels

The SEBS/PP ratio, the oil content filled in SEBS, and the amount of PB added were three crucial parameters of the primary formula. These parameters were considered as the factors of the orthogonal experiment and were defined as factor A, factor B, and factor C, respectively. The specific factor levels and experimental schemes are shown in Table 1 and Table 2, respectively.

#### 3.1.2. Orthogonal Test and Performance Evaluation

In the design of the orthogonal experiment (DOE), the values of K_I_, K_II_, K_III_, and K_IV_ value were, respectively, calculated by the sum of all experiment results in level 1, 2, 3, and 4 of a factor. K_Average_ value was calculated by the average of K_I_, K_II_, K_III_, and K_IV_ of a factor. The R-value, which is used to characterize the degree of influence of a factor, was calculated by the difference between the maximum minus the minimum in K_I_, K_II_, K_III_, and K_IV_ divided by K. Higher R value of DOE result means greater influence degree.

We carried out data analysis and present the results in Table 3, and we calculated the K_Average_ value and R value of hardness and each factor. The results are shown in Table 4.

According to Table 4, with the change of factor A, the hardness changed most obviously, which indicates the factor A (the ratio of SEBS/PP) had the greatest influence on the hardness of TPE-S (RA = 16.3). While the proportion of PP increased, the hardness increased significantly. The effect of the oil content added to SEBS on the TPE-S hardness was far less significant compared to the proportion of PP. The hardness of TPE-S decreased linearly with the increase in oil ratio filled in SEBS when the ratio of paraffin oil/SEBS was below 100/100. However, when the ratio exceeded 100/100, the influence of oil content on the hardness of TPE-S was reduced. PB-1 had a lower hardness than PP, and when PP was replaced by PB-1 in a certain proportion, the hardness of TPE-S decreased. It can be seen from Figure 1 that, in the formula, PP/SEBS ratio exhibited the greatest influence on the hardness of final TPE-S. When the ratio of paraffin oil/SEBS was within 100/100, the oil content filled in SEBS had a greater effect on hardness than the ratio of PB-1-replaced PP. When the ratio of paraffin oil/SEBS was higher than 100/100, the effect of oil content filled in SEBS on hardness was basically similar to the ratio of PB-1 replaced PP on hardness.

According to Table 3, the influence of each factor on the MFR was calculated by K_Average_ value and R value, and the results are shown in Table 5.

According to Table 5, the factor A (RA = 44.15), which was the ratio of SEBS/PP, significantly impacted MFR. It was speculated that this could be due to the use of ultra-high flow grade PP in the formulation, which had an MFR of over 1000 g/10 min—around 10 times higher than the final TPE, which had an MFR between 70 and 150 g/10 min. Therefore, increasing the amount of PP significantly improved the MFR of TPE. Furthermore, the ratio of paraffin oil filled in SEBS also had a substantial impact on the fluidity of the final TPE. Its effect was only surpassed by the addition ratio of ultra-high flow grade PP. As the oil content filled in SEBS increased, the MFR of TPE improved significantly. However, the fluidity of PB was low, and so, when the amount added to the system was less than 10%, adding a small amount of PB had little impact on the system. PB’s MFR was much lower than that of PP used in the formula, and its influence became more obvious when more than 10% of PP was replaced with PB. A higher proportion of ultra-high flow rate replace PB resulted in a decrease in the MFR of prepared TPE. The mechanical performance data of the material was analyzed, and the data are shown in Table 6, Table 7 and Table 8.

According to Table 6, Table 7 and Table 8 factor A had a greater influence on mechanical performance. As the PP content in SEBS/PP increased, the prepared TPE’s tensile strength and stress at 100% strain increased, while the elongation at break decreased slightly. In this material, the plastic phase PP provided rigidity and strength primarily. As the PP content increased, the tensile strength and the stress at 100% strain increased accordingly. However, the elongation at break decreased with an increase in PP content. Increasing the filling oil content was conducive to the disentanglement of SEBS molecular chains, conducive to the stretching of molecular chains, and the flexibility of molecular chains was improved, but the intermolecular force was also weakened, the tensile strength and fixed elongation were slightly reduced, and the elongation was slightly increased but not obvious. However, these changes were not significant. The addition of PB to the mixture led to poor compatibility with PP, resulting in decreased tear strength and elongation at break.

Table 9 demonstrates that tear strength was significantly influenced by Factor A, which refers to the SEBS/PP ratio. As the PP ratio increased, tear strength visibly improved. On the other hand, when oil content was added to SEBS, the mobility of SEBS molecular segment in the material increased, resulting in decreased compliance and lower tear strength.

According to Table 10, factor A (the ratio of SEBS/PP) had the greatest influence on friction and wear of final TPE. With an increase in PP content in the formula, the hardness and rigidity of the prepared TPE increased, resulting in improved wear resistance.

Moreover, an increase in PB-1 content slightly improved abrasion resistance, possibly due to PB-1’s inherent characteristic of good abrasion resistance. Additionally, the molecular chain of PB-1 contained butene units, which enabled better SEBS compatibility with PB-1 than with PP, thereby contributing to the abrasion resistance of the prepared TPE. As a result, the abrasion resistance of the prepared TPE improved with increased PB-1 content.

However, increasing the oil content filled in SEBS led to an increase in the surface stickiness degree of the prepared TPE, resulting in increased adhesive wear, leading to a decrease in overall wear resistance.

### 3.2. Influence of Oil Content Filled in SEBS on the Performance of TPE Skin Material

Because the conventional hardness of the automotive skin was within the range of 75 ± 5 A, formulas 1#, 2#, 6#, 7#, and 8# were suitable. The five experiments were compared and analyzed in Table 11.

It was found that 7# had the best wear resistance, followed by 8#. Because of high melt flow rate, 7# and 8# were more suitable for large-thin skins preparation. Combined with mechanical performance, the above five experiments all met the requirements and 7# was preferred. The SEM observation of 7# and 8# is shown in Figure 1.

From Figure 1a,b, it is evident that the abrasion scars on specimens 7# and 8# were not very distinct, while the concave-convex interface of the skin texture was quite clear. However, the abrasion scars of 8# were slightly more noticeable compared to 7#. Upon magnification to 200 times at the abrasion scar, as depicted in Figure 1c,d, it can be observed that the interface of the concave–convex skin texture at the abrasion scar of specimen 8# was partially obscured. In contrast, although specimen 7# exhibited partial wear, the interface remained lucid. Therefore, it can be inferred that the abrasion resistance of 7# surpassed that of 8#. It was tentatively speculated that the increase in adhesive wear and fatigue wear was attributable to the high oil content in 8#.

From the comparison of in Figure 2a,b, an interpenetrating polymer network with a co-continuous phase structure can be observed in AFM images of samples prepared from 7# and 8#. Distribution of the dispersed phase and continuous phase of 7# were more uniform, indicating better compatibility and interfacial bonding force, smoother surface, and a reduction in abrasion loss and improvement in fatigue wear resistance. Therefore, 7# exhibited better abrasion resistance.

### 3.3. Study on Compounding SEBS with Different Structures

#### 3.3.1. Influence of High Styrene Content SEBS/Low Styrene Content SEBS Ratio on Properties of TPE Skin Material

According to Table 12 and Figure 3, as the proportion of SEBS with high styrene content increased, the hardness of prepared TPES increased. The PS phase in SEBS was the hard segment, which provided hardness and rigidity. Therefore, as the styrene content increased, the hardness increased.

According to Table 13, the melt flow rate of prepared TPES decreased as the proportion of SEBS with high styrene content increased. Compared with the ethylene–butylene segment, the styrene segment in SEBS had higher steric hindrance. So, as the styrene content in SEBS increased, the melt flow rate of prepared TPES decreased.

In general, SEBS with high styrene content exhibited higher mechanical strength than that of SEBS with low styrene content. It can be seen from Table 13 that when the ratio of SEBS with high styrene content to SEBS with low styrene content was less than 30/70, the tensile strength of the material increased as the proportion of SEBS with high styrene content increased. However, when the ratio exceeded 30/70, the tensile strength gradually decreased as the proportion of SEBS with high styrene content increased. It was initially speculated that the benzene ring structure content in SEBS molecules increased, leading to a decrease in the ethylene–butene segment compatible with polypropylene content in SEBS, which resulted in worse compatibility between SEBS and polypropylene. This decrease in interfacial binding force between SEBS and PP caused vulnerability of the interface between the two materials under external forces, resulting in reduced tensile strength.

Polished version: A higher content of styrene in SEBS led to stronger intermolecular forces. As shown in Table 13, when the proportion of SEBS with high styrene content to SEBS with low styrene content was below 30/70, an increase in the proportion of SEBS with high styrene content increased the intermolecular forces within the SEBS molecules. This resulted in an increase in the stress at 100% strain. However, when the ratio of SEBS with high styrene content to SEBS with low styrene content exceeded 30/70, a further increase in the proportion of SEBS with high styrene content leads to a decrease in the stress at 100% strain. This was due to a decrease in the compatibility between SEBS and PP.

According to Table 13, as the proportion of SEBS with high styrene content increased, the elongation at break decreased, especially when the proportion exceeded 40%. The decrease in elongation at break was significant in this scenario. It was speculated that SEBS with high styrene phase had poor compatibility with PP, resulting in a decrease in elongation at break.

As can be seen from Figure 4 and Table 13, when the ratio of SEBS with low styrene content to SEBS with high styrene content was 80/20, 70/30, and 60/40, the appearance rating of the material was better, which was further compared with the wear test piece of the system with the ratio of 20/80.

It can be observed from Figure 5 that a clear interface was maintained even when the ratio of SEBS with low styrene content to SEBS with high styrene content was 80/20, 70/30, and 60/40. However, there were noticeable abrasion scars in the convex area of the grain. The best abrasion resistance was achieved when the ratio was 70/30, as shown in Figure 5b, where the interface remained intact and clear. On the contrary, when the ratio was 20/80, the content of benzene ring structure was high, implying a low content of ethylene–butene with excellent compatibility with PP, leading to weak compatibility between SEBS and polypropylene. Consequently, the interface bonding force between SEBS and PP was weak, resulting in surface abrasion resistance weakness of prepared TPES.

#### 3.3.2. Influence of Linear and Radial Compounding on Properties of TPE Skin Material

According to Table 14 and Figure 6, the MFR and hardness of prepared TPES increased in proportion to the increase in star-shaped SEBS content. It was suggested that in comparison to star-shaped SEBS with the same molecular weight, linear SEBS with a larger mean square terminal distance provided more physical entanglement points. Star-shaped SEBS contains more diblock SEBs, which act as plasticizers [13]. The prepared TPE made of star-shaped SEBS with the same molecular weight grade exhibited a slightly improved melt flow rate; as the proportion of star-shaped SEBS increased, the fluidity of the final TPES also increased. Moreover, the hardness of star-shaped SEBS was slightly higher than that of linear SEBS with a similar molecular weight and styrene content; hence, the hardness of the prepared TPE increased with the increase in star-shaped SEBS content.

Based on Figure 7, it can be observed that as the amount of star-shaped SEBS increased, the tensile strength and tear strength gradually decreased. This may be due to a potential decrease in the compatibility between SEBS and PP, resulting in a decline in mechanical properties.

According to Figure 8 and Figure 9, it can be observed that TPES with a linear/star SEBS ratio of 70/30 exhibited the best abrasion resistance. However, the abrasion resistance of the prepared TPE decreased as the proportion of star-shaped SEBS increases. This was evident from the gradual smoothing of the convex areas on the surface of the sample piece after abrasion, and the blurring of the boundaries of the concave-convex skin texture. A preliminary hypothesis was that the addition of radial SEBS increased the number of branched chains in the SEBS molecules, making it less likely for the molecular chain to slip, thereby increasing abrasion resistance.

Therefore, it can be concluded that the abrasion resistance of prepared TPE with a 70/30 linear/star SEBS ratio was better than that with an 80/20 ratio. However, the abrasion resistance of prepared TPES decreased as the ratio of star SEBS to linear SEBS further increased, such as to 60/40 or even higher. This was due to worse compatibility and weak interfacial force of star SEBS with linear SEBS and PP.

## 4. Conclusions

(1)Through orthogonal experiments, it was found that the ratio of SEBS/PP had the greatest impact on the mechanical properties, melt flow rate, and abrasion resistance of prepared TPES. The mechanical performance and abrasion resistance of the TPES improved with the appropriate increase in the PP content. However, because of the requirement of hardness between 70 A and 80 A for interior soft skin, too high content of polypropylene in the formula was also not suitable.(2)The amount of filling oil in the system should not be excessive. Increasing the oil content in SEBS results in an increase in the degree of stickiness on the surface of prepared TPE, leading to worsened overall wear resistance and increased adhesive wear behavior. Therefore, while ensuring a sufficient melt flow rate for interior skin injection process, filling too much oil content in SEBS is not recommended.(3)A better overall performance of prepared TPES can be achieved by combining SEBS with high styrene content and SEBS with low styrene content in a 30/70 ratio. However, if the ratio exceeds 30/70, an increase in SEBS with a high styrene ratio will lead to poorer compatibility and a decrease in the mechanical performance and abrasion resistance of the prepared TPES.(4)The varying proportions of linear/star SEBS also had a significant impact on the properties of TPES prepared. The best abrasion resistance and good mechanical performance were achieved when the ratio of linear/radial SEBS was 70/30.

## Figures and Tables

**Figure 1 polymers-15-02753-f001:**
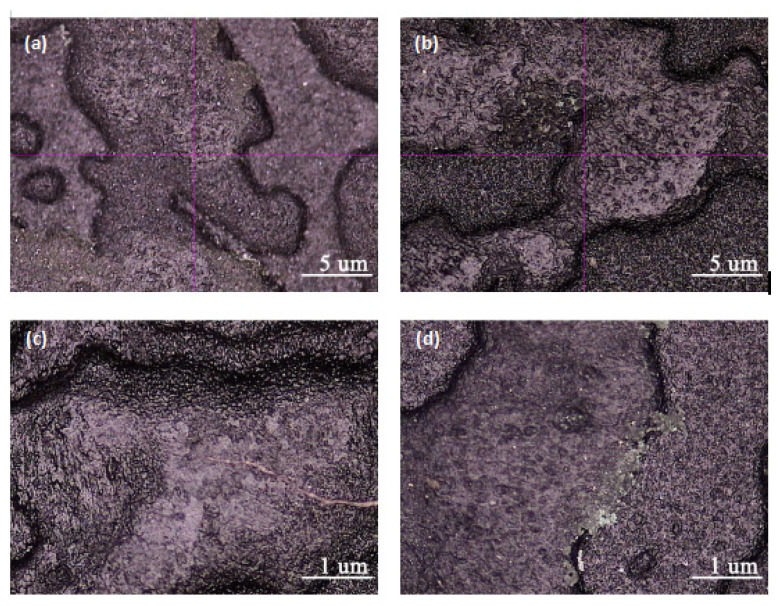
Comparison of abrasion scars of Sample 7 and Sample 8. (**a**,**b**) are the pictures of Sample 7 and Sample 8 magnified 40 times, respectively, (**c**,**d**) are partial magnifications of (**a**,**b**) with a magnification of 200 times.

**Figure 2 polymers-15-02753-f002:**
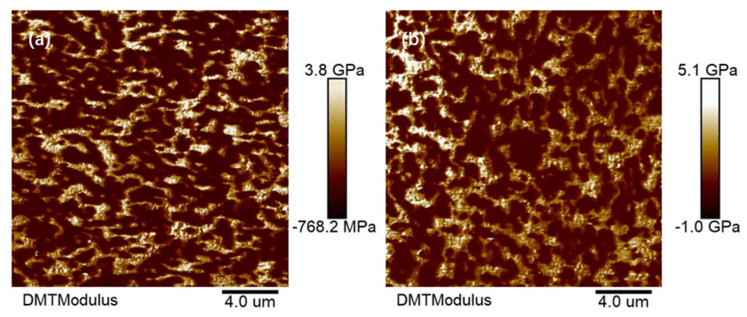
AFM Images of Sample 7 and Sample 8. (**a**) Sample 7, (**b**) Sample 8.

**Figure 3 polymers-15-02753-f003:**
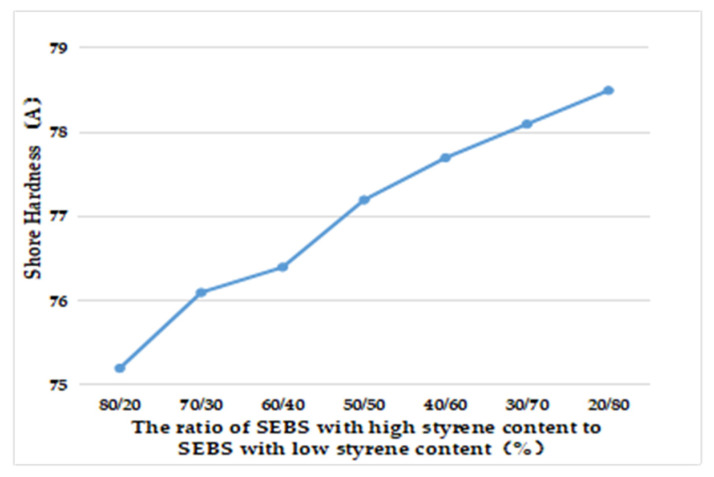
Effect of the ratio of high styrene content to low styrene content of SEBS on Shore Hardness.

**Figure 4 polymers-15-02753-f004:**
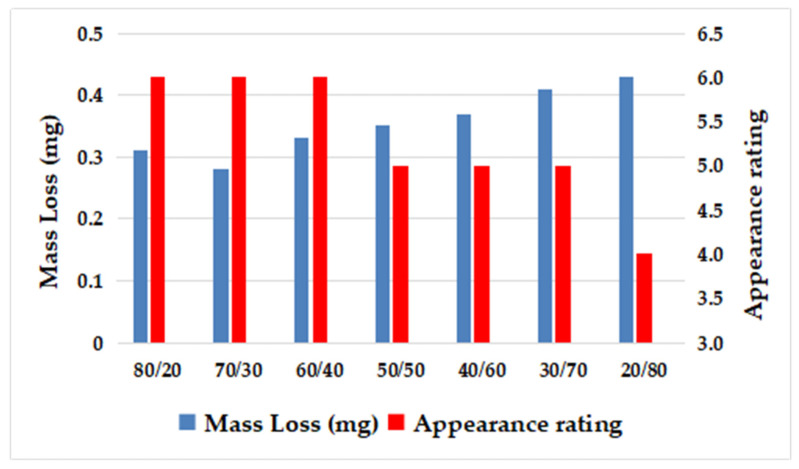
Effect of the ratio of high styrene content to low styrene content of SEBS on Taber Abrasion.

**Figure 5 polymers-15-02753-f005:**
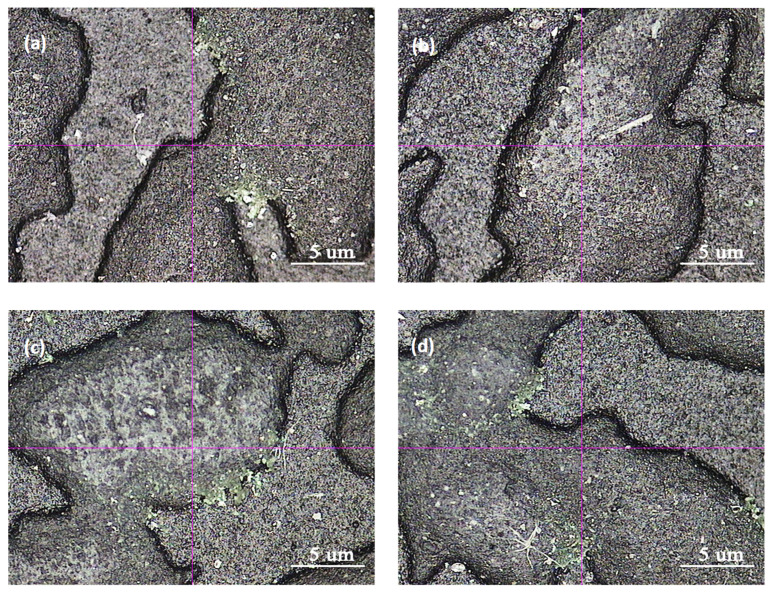
Comparison of abrasion scars of sample of SEBS with different styrene content on Taber Abrasion. (**a**) 80/20, (**b**) 70/30, (**c**) 60/40, (**d**) 20/80.

**Figure 6 polymers-15-02753-f006:**
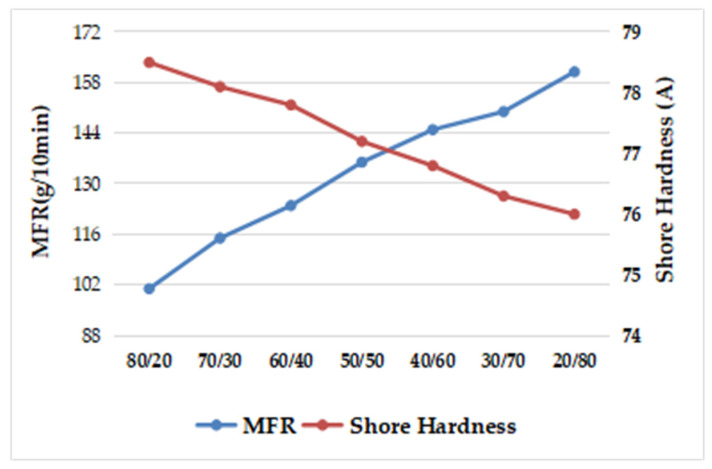
Effect of SEBS with different ratio of star SEBS on MFR and Hardness.

**Figure 7 polymers-15-02753-f007:**
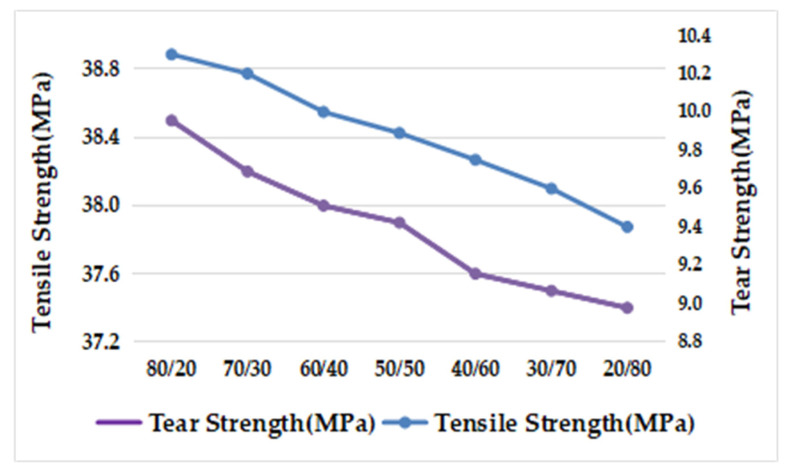
Effect of SEBS with different ratio of radial SEBS on Tear Strength and Tensile Strength.

**Figure 8 polymers-15-02753-f008:**
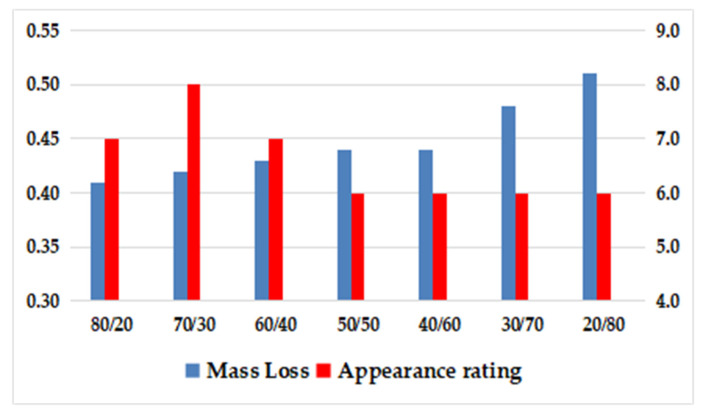
Effect of SEBS with different ratio of radial SEBS on Taber Abrasion.

**Figure 9 polymers-15-02753-f009:**
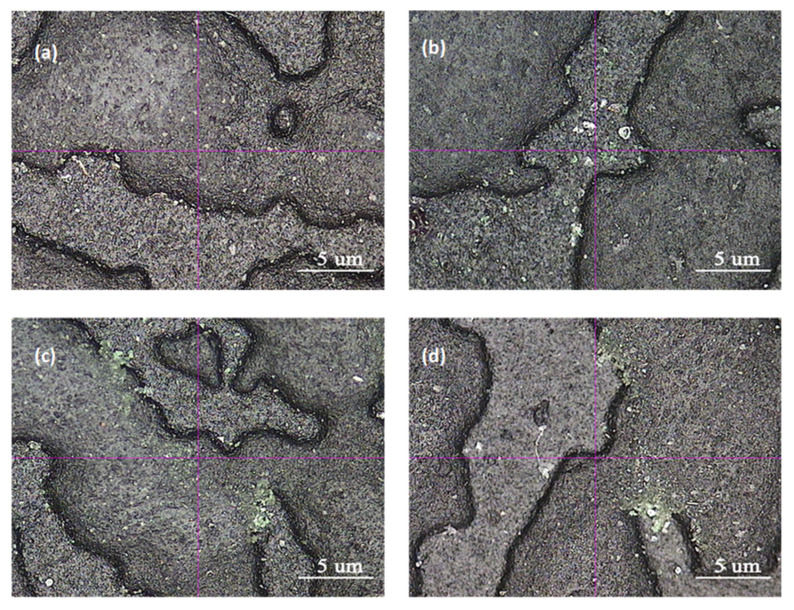
Comparison of abrasion scars of sample of SEBS with different ratio of radial SEBS on Taber Abrasion. (**a**) 80/20, (**b**) 70/30, (**c**) 60/40, (**d**) 50/50, (**e**) 40/60, (**f**) 30/70, (**g**) 20/70.

**Table 1 polymers-15-02753-t001:** Factors and levels of orthogonal experiment design.

	Factors	A (SEBS/PP)	B (Ratio of SEBS to Oil)	C (PB-1 Content in PP)
Level				
1		A1 (80/20)	B1 (100/50)	C1 (5%)
2		A2 (70/30)	B2 (100/75)	C2 (10%)
3		A3 (60/40)	B3 (100/100)	C3 (15%)
4		A4 (50/50)	B4 (100/125)	C4 (25%)

**Table 2 polymers-15-02753-t002:** L12 (34) orthogonal experiment design.

Number	Factors		
	A	B	C
1#	A1	B1	C1
2#	A1	B2	C2
3#	A1	B3	C3
4#	A1	B4	C4
5#	A2	B1	C2
6#	A2	B2	C1
7#	A2	B3	C4
8#	A2	B4	C3
9#	A3	B1	C3
10#	A3	B2	C4
11#	A3	B3	C1
12#	A3	B4	C2
13#	A4	B1	C4
14#	A4	B2	C3
15#	A4	B3	C2
16#	A4	B4	C1

**Table 3 polymers-15-02753-t003:** Mechanical properties of orthogonal experiment design.

No.	Shore Hardness (A)	MFR (g/10 min)	Tensile Strength (MPa)	Stress at 100% Strain (MPa)	Elongation at Break (%)	Tear Strength (KN/m)	Taber Abrasion	
							Mass Loss (mg)	Appearance Rating
1#	74.3	50.9	14.6	2.32	756	53.1	0.52	4
2#	73.5	70.5	13.9	2.26	777	48.4	0.41	4
3#	69.6	84	13.4	2.15	820	44.2	0.3	4
4#	65.8	90	11.4	2.07	833	36.9	0.23	5
5#	81.9	81	16.7	3.86	821	58.3	0.28	4
6#	77.9	95	14.0	3.42	800	57.7	0.28	4
7#	76.9	92	12.7	3.23	738	56.4	0.13	6
8#	75.5	102	11.6	3.05	702	54.0	0.19	6
9#	85.7	86	22.5	4.92	731	70.3	0.21	6
10#	85.0	93	22.3	4.86	688	69.3	0.19	6
11#	84.2	125	21.2	4.41	730	72.0	0.21	5
12#	83.9	127	19.8	4.60	717	72.4	0.2	6
13#	88.9	89	22.6	6.82	691	84.0	0.19	6
14#	88.3	110	21.5	6.92	729	86.1	0.18	6
15#	86.8	129	20.3	6.34	746	78.9	0.21	5
16#	84.4	144	19.0	5.49	757	63.8	0.25	5

**Table 4 polymers-15-02753-t004:** K_Average_ value and R value of hardness, the influence of various factors on hardness.

Shore Hardness	A	B	C
K_I_	283.2	330.8	320.8
K_II_	312.2	324.7	324
K_III_	338.8	317.5	319.1
K_IV_	348.4	309.6	316.6
K_Average_	320.65	320.65	320.125
R	16.3	5.3	1.85

**Table 5 polymers-15-02753-t005:** Calculation of various factors on MFR (K_Average_ value and R value).

MFR	A	B	C
K_I_	295	306	414
K_II_	370	368.5	407
K_III_	431	430	382
K_IV_	472	463	364
K_Average_	392.1	392.1	392.1
R	44.15	39.03	12.73

**Table 6 polymers-15-02753-t006:** Calculation of K_Average_ value and R value of each factor on Tensile Strength.

Tensile Strength	A	B	C
K_I_	53.3	76.4	68.8
K_II_	55	71.7	70.7
K_III_	85.8	67.6	69
K_IV_	83.4	61.8	69
K_Average_	69.375	69.375	69.375
R	8.125	3.65	0.475

**Table 7 polymers-15-02753-t007:** Calculation of K_Average_ value and R value of each factor on Stress at 100% Strain.

Stress at 100% Strain	A	B	C
K_I_	8.8	17.46	15.64
K_II_	13.56	17.46	17.06
K_III_	18.79	16.13	17.04
K_IV_	25.57	15.21	16.98
K_Average_	16.68	16.565	16.68
R	4.1925	0.5625	0.355

**Table 8 polymers-15-02753-t008:** Calculation of K_Average_ value and R value of each factor on Elongation at Break.

Elongation at Break	A	B	C
K_I_	3186	2999	3043
K_II_	3061	2994	3061
K_III_	2866	3034	2982
K_IV_	2923	3009	2950
K_Average_	3009	3009	3009
R	80	10	27.75

**Table 9 polymers-15-02753-t009:** Effects of different factors on Tear Strength.

Tear Strength	A	B	C
K_I_	182.6	265.7	246.6
K_II_	226.4	261.5	258
K_III_	284	251.5	254.6
K_IV_	312.8	227.1	246.6
K_Average_	251.45	251.45	251.45
R	32.55	9.65	2.85

**Table 10 polymers-15-02753-t010:** Effect of different factors on friction and wear.

Mass Loss	A	B	C
K_I_	1.46	1.2	1.26
K_II_	0.88	1.06	1.1
K_III_	0.81	0.85	0.88
K_IV_	0.83	0.87	0.74
K_Average_	0.995	0.995	0.995
R	0.65	0.35	0.52

**Table 11 polymers-15-02753-t011:** Mechanical properties of materials within the hardness range of 75 ± 5 A.

No.	Shore Hardness (A)	MFR (g/10 min)	Tensile Strength (MPa)	Stress at 100% Strain (MPa)	Elongation at Break (%)	Tear Strength (KN/m)	Taber Abrasion	
							Mass Loss (mg)	Appearance Rating
1#	74.3	50.9	14.6	2.32	756	53.1	0.52	4
2#	73.5	70.5	13.9	2.26	777	48.4	0.41	4
6#	77.9	95	14	3.42	800	57.7	0.28	4
7#	76.9	92	12.7	3.23	738	56.4	0.13	6
8#	75.5	102	11.6	3.05	702	54	0.19	6

**Table 12 polymers-15-02753-t012:** Different ratios of SEBS with high and low styrene content.

SEBS	Content of Styrene (%)	A	B	C	D	E	F	G
SEBS1	31%	80	70	60	50	40	30	20
SEBS3	56%	20	30	40	50	60	70	80

**Table 13 polymers-15-02753-t013:** Performance of different ratios of SEBS with high and low styrene content.

Ratio	80/20	70/30	60/40	50/50	40/60	30/70	20/80
MFR	100	85	75	70	67	65	60
Tensile Strength	11.4	12.1	11.3	10.3	9.8	9.1	8.8
Elongation at Break	765	760	758	741	718	695	677
Stress at 100% Strain	2.71	2.89	2.83	2.81	2.78	2.74	2.71
Mass Loss	0.31	0.28	0.33	0.35	0.37	0.41	0.43
Appearance rating	6	6	6	5	5	5	4

**Table 14 polymers-15-02753-t014:** Formula design of TPE prepared by SEBS with different molecular chain structures.

SEBS	Structural Type of the Molecule	1#	2#	3#	4#	5#	6#	7#
11651	Linear	880	770	660	550	440	330	220
YYH-602T	Star	220	330	440	550	660	770	880

## Data Availability

The data presented in this study are available on request from the corresponding author.

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
