# Peer review of "Effect of SEBS Molecular Structure and Formula Composition on the Performance of SEBS/PP TPE for Automotive Interior Skin"

_polymers, 2023, doi:10.3390/polym15122753_

Round 1
Reviewer 1 Report
I have read the manuscript entitled “Effect of SEBS Molecular Structure and Formula Composition on the Performance of SEBS/PP TPE for Automotive Interior Skin” and analyzed its potential for publication in the MDPI journal Polymers (ISSN 2073-4360).
In my opinion, the manuscript is interesting. The Introduction provides with decent level of background information. However, I have noticed two problems to solve.
1. The scientific background is not stressed clearly enough.
2. The objectives of the study are missing the highlighting of the novelty
After consideration I have got to the conclusion that the manuscript could be accepted for publication after minor revision.
Author Response
Thank you very much for your evaluation on our manuscript “Effect of SEBS Molecular Structure and Formula Composition on the Performance of SEBS/PP TPE for Automotive Interior Skin” and kind comments and suggestions. We have carefully read the comments. The old manuscript has been revised based on the comments received point by point. The English grammar, spelling, and sentence structure have been double checked through the whole manuscript. Attached please find the file for the new one. The revision parts of the manuscript have been marked in red or blue. The responses as following:
1. The scientific background is not stressed clearly enough.
Response: Good question! In the background part, it has been further enriched, hoping to better express the innovation of this paper. For details, see the modified part marked in the attachment.
2. The objectives of the study are missing the highlighting of the novelty.
Response: Good suggestion! Few people have studied and find the method to belance the between high fluidity and enough wear resistance and mechanical properties at present, which is very difficult but significant for interior skins prepared by injection molding. Therefore DOE method was apllied to research and try to find the balance. I think This is the real novelty of this manuscript. We have added the description in Line 81-92 of the revised version of the manuscript.

Reviewer 2 Report
Dear Authors
My opinion is that the paper needs a major revision to be published. Thank you for giving me the opportunity to review papers for Polymers.
Review:
Title: Effect of SEBS Molecular Structure and Formula Composition on the Performance of SEBS/PP TPE for Automotive Interior Skin
· The work has too many images, i.e. it is not necessary to have pictures and tables in the paper with the same results. Maybe it wouldn't be a bad thing to transfer one of those two to additional material.
· Line 117- It should be emphasized how the K value and R value were calculated.
· Line 122-134: one should also refer to some literature with similar results. Also in the paragraph line 144-155 there is no reference. In the rest of the text where the results are also not a single reference. On what basis do you compare your results and how do you know they are good when you don't have a single reference in the results and discussion section?
· In Table 4. - What do the marks I, II, III, and IV represent and what do these hardness values in the table represent?
· Sample labels are not clear, and difficult to follow through the text.
· Check the values in Table 3, tensile strength and elongation do not match.
Author Response
Thank you very much for your evaluation on our manuscript “Effect of SEBS Molecular Structure and Formula Composition on the Performance of SEBS/PP TPE for Automotive Interior Skin” and kind comments and suggestions. We have carefully read the comments. The old manuscript has been revised based on the comments received point by point. The English grammar, spelling, and sentence structure have been double checked through the whole manuscript. Attached please find the file for the new one. The revising traces was still left in the revised vesion of the manuscript. The responses to reviewers are as following:
1. The work has too many images, i.e. it is not necessary to have pictures and tables in the paper with the same results. Maybe it wouldn't be a bad thing to transfer one of those two to additional material.
Response: Good suggestion! At present, the duplicate pictures have been deleted from the manuscript, and the modify traces can be find in the revised version. Thank you very much for your valuable suggestion!
2. Line 117- It should be emphasized how the K value and R value were calculated.
Response: Good suggestion! Both K value and R value are commonly used data in orthogonal experimental analysis with standard calculated methods. For example, in this manuscript I-value or KI-value is calculated by the sum of all experiment results in level 1 of a factor, and K-value is calculated by the average of KI, KII, KIII and KIV of a factor. In order to be easily understood, the original K in the article has been changed to KAverage. The R-value is calculated by the difference between the maximum minus the minimum in KI, KII, KIII and KIV divided by K. Because the two calculations are standard method and well known in DOE experiments, the numerical values are given directly in the paper without explanation in old version. Now means and calculated methods of R value and KAverage has also been described in line 122-127 of revised version.
3. Line 122-134:one should also refer to some literature with similar results. Also in the paragraph line 144-155 : there is no reference. In the rest of the text where the results are also not a single reference. On what basis do you compare your results and how do you know they are good when you don't have a single reference in the results and discussion section?
Response: DOE experiment is not a method for directly finding formula with good or best performance but a method for studying the influence degree of the factors to particular performance and identify main influence factors, which will be very important guidance for Formula and properties design of materials, R value is used to characterize the degree of influence of a factor, Higher R value of DOE result means greater influence degree.
4. In Table 4. - What do the marks I, II, III, and IV represent and what do these hardness values in the table represent?
Response: Good question! The I, II, III, IV, K and R are all standard representations of DOE experiment. For example, I-value is calculated by the sum of all experiment results in level 1. K is calculated by the average of I, II, III and IV. R is calculated by the difference between the maximum minus the minimum in I, II, III and IV divided by K. For being easily understood, I, II, III, IV and K have been changed to Kâ… ,Kâ…¡,,K III, KIV and KAverage in revised version, which have the same representations. Means and calculated methods of KI, KII, KIII, KIV, R value and KAverage value has also been described in line 122-127 of revised version.
5. Sample labels are not clear, and difficult to follow through the text.
Response: Good question! The numbers in the table have been rearranged for better understanding.
6. Check the values in Table 3, tensile strength and elongation do not match.
Response: Sorry, we have rechecked Table 3 and can not find mismatch values. The mismatch may be caused by word software with different version of the computer between yours and mine. would you mind to help me to check the revised version and let me know if the problem is still exists ? Thank you very much!

Round 2
Reviewer 2 Report
Dear Authors,
the paper can be accepted in its current form.
Best regards,
Dr Marija Vuksanovic